# Influence of Leguminous Cover Crops on Soil Chemical and Biological Properties in a No-Till Tropical Fruit Orchard

Ariel Freidenreich [1,2] , Sanku Dattamudi [1] , Yuncong Li [2,3] and Krishnaswamy Jayachandran [1,*]

1 Department of Earth and Environment, Florida International University, Miami, FL 33199, USA; afreidenreich@ufl.edu (A.F.); sdattamu@fiu.edu (S.D.)
2 Department of Soil, Water, and Ecosystem Sciences, University of Florida, Gainesville, FL 32611, USA; yunli@ufl.edu
3 Tropical Research and Education Center, University of Florida, Homestead, FL 33031, USA
* Correspondence: jayachan@fiu.edu; Tel.: +1-(305)-348-6553

**Abstract:** South Florida's agricultural soils are traditionally low in organic matter (OM) and high in carbonate rock fragments. These calcareous soils are inherently nutrient-poor and require management for successful crop production. Sunn hemp (SH, *Crotalaria juncea*) and velvet bean (VB, *Mucuna pruriens*) are highly productive leguminous cover crops (CCs) that have shown potential to add large quantities of dry biomass to nutrient- and organic-matter-limited systems. This study focuses on intercropping these two CCs with young carambola (*Averrhoa carambola*) trees. The objective was to test the effectiveness of green manure crops in providing nutrients and supplementing traditional fertilizer regimes with a sustainable soil-building option. Typically, poultry manure (PM) is the standard fertilizer used in organic or sustainable production in the study area. As such, PM treatments and fallow were included for comparison. The treatments were fallow control (F), fallow with PM (FM), sunn hemp (SH), SH with PM (SHM), velvet bean (VB), and VB with PM (VBM). Sunn hemp and VB were grown for two summer growing seasons. At the end of each 90-day growing period, the CCs were terminated and left on the soil surface to decompose in a no-till fashion. The results suggest that SH treatments produced the greatest amount of dry biomass material ranging from 48 to 71% higher than VB over two growing seasons. As a result, SH CCs also accumulated significantly higher amounts of total carbon (TC) and total nitrogen (TN) within their dry biomass that was added to the soil. Sunn hemp, SHM, and FM treatments showed the greatest accumulation of soil OM, TC, and TN. Soil inorganic N ($NH_4^+ + NO_3^- + NO_2$) fluctuated throughout the experiment. Our results indicate that generally, VB-treated soils had their highest available N around 2 months post termination, while SH-treated soils exhibited significantly higher N values at CC termination time. Sunn hemp + PM (SHM) treatments had highest soil N availability around 4 months after CC termination. Soil enzyme activity results indicate that at CC termination, SHM exhibited the highest levels of β-1-4- glucosidase and β-N-acetylglucosaminidase among all treatments. Overall, SH, SHM, and FM treatments showed the greatest potential for supplementing soil nutrients and organic matter in a no-till fruit production setting.

**Keywords:** sunn hemp (*Crotalaria juncea*); velvet bean (*Mucuna pruriens*); carambola (*Averrhoa carambola*); soil health; soil enzyme activity

## 1. Introduction

Conservation agriculture, a sustainable approach to maximize crop production while preserving environmental quality, has become an increasingly popular subject of research. This is largely attributable to the lack of applicable information regarding sustainable agriculture components and their harmony in organic agroecosystems. Cover cropping is a widely recognized conservation agricultural strategy [1–3], specifically for low-input agricultural systems. The soil ecosystem services provided by cover crops are substantial,

specifically in the form of organic carbon additions [4]. No-till (NT) farming, combined with cover cropping, is often considered as an ideal system for judicious resource utilization in maximizing return on investment (ROI) and maintaining soil biodiversity [5]. Fruit orchards are traditionally NT systems once trees are planted, as mechanical tillage would cause damage to surface feeder roots that remain in the top centimeters of the soil. Therefore, soil surrounding perennial tree crops is often left unstimulated, leaving trees to ultimately experience a reduction in productivity, as crop rotation and tillage is not possible in most cases [6]. Perennial NT systems can significantly benefit from cover crop species that produce large amounts of dry biomass to achieve high C inputs to soil [7].

In tropical fruit production settings, farmers face various challenges which stem from land management in warm and wet climates. This phenomenon is reflected through quick decomposition of organic matter [8–10] and higher pressure from pests and diseases [11]. Leguminous cover crops, specifically varieties suited for tropical climates, have great potential to ameliorate these issues by improving soil resilience and enhancing farmland diversity [12]. Sunn hemp (SH, *Crotalaria juncea*) and velvet bean (VB, *Mucuna pruriens*), two commonly used leguminous cover crops, have been shown to fix 40 to 80 kg N ha$^{-1}$ in tropical climates [13,14]. No-till organic ecosystems can significantly benefit from cover crop species such as SH and VB which produce large amounts of dry biomass to achieve high C inputs to soil [7]. Previous studies found that SH and VB can significantly increase soil C and N fractions, improve soil aggregate stability, and influence the abundance of beneficial soil microbes in no-till tropical production [15–18]). Consequently, it is of great importance to study soil nutrient cycling in harmony with soil microbiota. Microbial functional diversity is a driver for a plethora of ecological and environmental interactions [19]. Soil enzymes are directly related to soil microbial activity and overall soil fertility [20], making them a dynamic indicator for the effectiveness of soil amendments in stimulating nutrient cycling and mineralization rates.

The popularity of organic agriculture has increased more than 550% worldwide within the last couple of decades [21]. Consumers are more interested in 'healthy food habits' and relate that concept to products coming from organic farms. Although organic consumables are becoming increasingly more available, sustainable production of organic commodities can be challenging. Additionally, due to rapid integration of organic farming into large commercial food-production systems, small and medium-size organic growers are struggling to make a minimum profit. Certified organic farms in the US saw a ~39% increase from 2012 to 2017, while Florida's organic farms only increased by ~4.5% within the same time frame [22]. However, the numbers of small and mid-size farmers in South Florida are decreasing, and currently, fewer than ten certified organic vegetable growers can be found in Miami–Dade [23], the major fruit- and vegetable-producing county in South Florida. In addition, soils in South Florida are predominantly porous sandy loam with very low organic matter content (less than 2%, [24]), which often causes production problems for the local growers.

Cover-cropping practices combined with low-cost organic nutrient sources could potentially help small and mid-size farmers achieve better economic return and promote environmental and economic sustainability. Composted poultry manure (PM) is commonly applied as a fertilizer in organic production systems [25,26]. Nutrients such as C and N in poultry manure are in organic forms [27,28] and are released slowly as the materials decompose. This process is similar to that employed in commercially available slow-release fertilizers that have become standard for reducing nutrient leaching and protecting water quality.

Carambola (*Averrhoa carambola*), more commonly known as starfruit, is a tropical fruit tree native to Southeast Asia. The carambola tree is small to medium in height and produces fruit mainly in the mid-canopy area. Carambola is accustomed to hot, humid weather, making it ideal for growth in (sub)tropical climates, and consequently, South Florida is the only location in the contiguous US where carambola is produced commercially [29]. Carambola production has been estimated to contribute ~$3.7 million

to Florida's economy, most of which comes from South Florida [30]. Carambola has huge potential for South Florida growers as a lucrative cash crop. For many years, the avocado has been the staple tropical fruit crop for Miami–Dade County growers (over 6000 planted acres, [31]). However, within the last decade, the aggressive emergence of a devastating fungal pathogen, commonly known as laurel wilt (*Raffaelea lauricola*), has led to the mandatory eradication of many infected avocado groves [32–34]. As such, many local growers are looking towards alternative tropical crops to populate their groves. Carambola is a promising candidate, as its current individual tree value is highest for all maturity increments (1–3 years: $567, 4–6 years: $860, 7+ years: $984) as compared to avocado and other feasible alternatives [35].

In an effort to target current concerns within the Miami–Dade County agricultural scope and to explore solutions to improve management practices for sustainable production of tropical fruits worldwide, we developed a study to test the effectiveness of SH and VB as cover crops. The goal was to quantify the response of dynamic soil characteristics to cover crop incorporation in a young carambola grove by exploring responsive soil parameters. The specific objectives of this 2-season study were to (1) assess carbon and nitrogen inputs from cover crops and poultry manure incorporated into an organic carambola grove, (2) monitor physiochemical soil responses to these inputs in a no-till setting, and (3) assess soil enzymatic activity in response to these added amendments.

## 2. Materials and Methods

### 2.1. Site Location and Characteristics

This 2-season (May 2018 to December 2019) field experiment took place in a certified organic (as listed by USDA-AMS) fruit orchard (6.07 ha) located in the Redlands Agricultural Area (RAA) of South Florida, United States. The RAA is subtropical and located in plant hardiness zone 10b [36] (USDA, 2012). This subtropical climate is characterized by a typical wet summer season (May–October, 26.6 °C average temperature, and 18 cm average annual rainfall) and a dry winter season (November–April, 21.1 °C average temperature, 4.6 cm average annual rainfall) with warm weather year-round (23.6 °C average temperature) [37].

The USDA NRCS National Cooperative Soil Survey categorizes the soil in the RAA as Krome series soil, high in calcium carbonate ($CaCO_3$) rock fragments and generally recognized as gravelly loam [38]. The soil profile is shallow, with a plowed layer ranging from 0 to 18 cm. The limestone parent material has resulted in well-drained and slightly alkaline soil. As a result of this shallow soil profile, rock plowing is a common practice in agricultural fields to create enough depth (10–20 cm) for root growth and establishment. In addition to rock plowing, tropical fruit managers typically trench their land (46 to 61 cm deep and 41 to 46 cm wide) to ensure enough depth for tree root growth and anchoring for protection during tropical cyclones [39].

### 2.2. Experimental Design

One year before the start of the experiment, two rows of young carambola trees were planted, extending 122 m long with 7 m spacing between rows (Figure 1). The trees used for this study were ~three-year-old 'Hawaiian Super Sweet' trees grafted onto 'Golden Star' seedling rootstocks. Carambola saplings were planted with 3.8 m between each tree, resulting in 30 trees per row, and 54 trees were randomly selected for treatment. Experimental sites were arranged in a completely randomized design (CRD) with two cover crop treatments: sunn hemp (SH) and velvet bean (VB); two cover crop + manure treatments: sunn hemp + poultry manure (SHM) and velvet bean + poultry manure (VBM); and two fallow control treatments: fallow (F) and fallow + poultry manure (FM). The design included six treatments with nine replications for each treatment, involving a total of 54 trees. Twenty-seven trees were treated with an organic composted fertilizer amendment (5N-3P-2K USDA Organic Certified poultry manure), and the other 27 trees did not receive any fertilizer treatments. Details about the experimental timeframe and treatments can be found in our other published work [40].

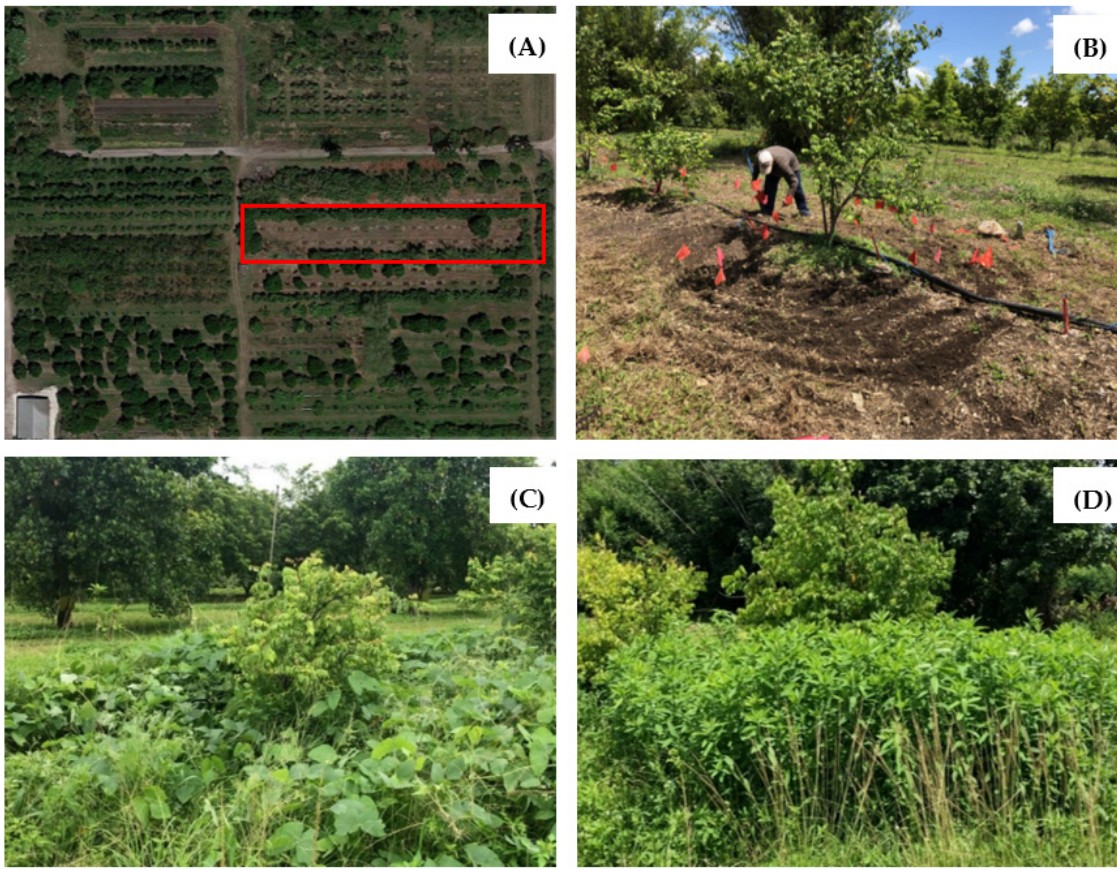

**Figure 1.** (**A**) The location of the experiment was a multi-use tropical fruit grove (experimental area highlighted in red), (**B**) seeding cover crops around carambola trees in a circular fashion, (**C**) velvet bean established in experimental plots, (**D**) sunn hemp established in experimental plots.

### 2.3. Field Sampling and Laboratory Analyses

2.3.1. Field Methodology

Weeds were physically removed from each plot before the start of the experiment in preparation for cover crop seeding. A planting area of 8.8 m² was established, and CCs were seeded directly in a concentric pattern starting at the dripline (approx. 0.5 m radius from the trunk) and circling around the tree in a 1.25 m radius (Figure 1). Carambola trees treated with cover crops received either 33 kg ha$^{-1}$ (89 g/plot) of SH (*Crotalaria juncea* L. cv. 'Tropic Sun') seed or 25 kg ha$^{-1}$ (67 g/plot) VB (*Mucuna pruriens* var. *'pruriens'*) seed. Seeding rates were calculated following the Miami–Dade County Extension recommendations for CC seeding in vegetable crop scenarios and adjusted to a 33% grove coverage rate. The CC coverage used for this study was determined based on size of trees, spacing, and management equipment. Cover crop seeds were treated with OMRI-certified Guard'n Seed Inoculant (Verdesian Life Sciences, Cary, NC, USA), which contains a variety of rhizobium species (*Bradyrhizobium japonicum*, *Bradyrhizobium sp. (Vigna)*, *Rhizobium leguminorsarum biovar viceae*, and *Rhizobium leguminosarum biovar phaseoli*), to facilitate root nodulation and N fixation.

Sunn hemp and VB treatments were planted simultaneously. All plots were irrigated for one hour per day via sprinkler system. Sunn hemp and VB were terminated 90 days after germination. Sunn hemp was terminated mechanically via hedge trimmer, and VB was hand clipped, leaving root systems intact. The cover crop biomass was laid around the base of each respective tree to decompose on the soil surface. Following termination, fertilizer treatments (1.4 kg poultry manure per tree) were applied every two months except for the CC growing season (4 times per year), resulting in 120 kg ha$^{-1}$ N added per year from PM, as per recommendation by Crane, 2001 [41]. The carbon content of the poultry

manure was $33.76 \pm 0.29\%$, and the N content was $5.53 \pm 0.13\%$. The cover crop treatments were planted and terminated two times over this study period. Individual treatments for each tree remained the same for both years.

Composite soil samples (0–15 cm depth, four per plot per sampling time) were collected over the course of the experiment from the planting area of each tree. Soil samples were taken before cover crop planting, before termination, and every 2 months following, except for the last 4 months in which sampling occurred once per month. Temperature, moisture percentage, and electrical conductivity (EC) (STEVENS Hydraprobe, Portland, OR, USA) were measured once per month and at corresponding soil-sampling times. At 90 days after cover crop seed germination, a 40 cm$^2$ area of plant matter (cover crop and weed) was collected from each plot, including control plots. Aboveground biomass was measured to determine organic matter and nutrient additions to the soil.

### 2.3.2. Soil Physicochemical Properties

Composite soil samples were oven-dried (30 °C for 72 h), sieved (2 mm), and ground to prepare for analysis of chemical properties. Soil organic matter (SOM) was determined through the standard loss-on-ignition method (550 °C for 4 h). Soil total carbon and nitrogen (CN) were measured via dry combustion using a Truspec Carbon/Nitrogen analyzer (LECO Corporation, St. Joseph, MI, USA).

Inorganic N was extracted using a 2M KCl extraction method. Extracts were then analyzed for nitrate ($NO_3^-$) and nitrite ($NO_2^-$) following USEPA Nitrate-Nitrite by Automated Colorimetry Method 353.2, Revision 2.0 (1993) [42]. These same extracts were used for ammonium ($NH_4^+$) determination following USEPA Method 350.1, Revision 2.0 (1993) [43]. All readings were quantified with a SEAL Analytical AQ2 Discrete Auto Analyzer (Mequon, WI, USA).

Soil moisture content was determined via the gravimetric method (dried at 105 °C for 24 h) and bulk density via cylinder method. Soil textural analysis was performed using the standard hydrometer method. All analysis was conducted at Florida International University within the Soil–Plant–Microbiology Laboratories (Miami, FL, USA).

### 2.3.3. Soil Enzyme Activity

Soil enzyme analysis was conducted for determination of β-1-4-glucosidase (C) and β-N-acetylglucosaminidase (N). A methodology adopted from Sinsabaugh et al. (1997), Hoppe (1993), and Chróst and Kambeck (1986) [44–46] was utilized to determine soil enzyme activity using differences in concentration of fluorescent substrate released during incubation time compared with no incubation. Soil slurries with a 2:1 water-to-soil ratio (4 g distilled deionized water to 2 g fresh soil) were made, and pH readings were taken. Substrates were prepared using morpholinoethanesulfonic acid (MES) in combination with 4-methylumbelliferone (MUF) β-D-glucosidase (MUF-C) and MUF-N-acetyl- β-D-glucosaminide (MUF-N). Soil floc was prepared at varying dilutions according to concentration. For C and N, $10^{-2}$ dilutions were analyzed using a Synergy HT Multi-Mode 96 Well Plate Reader (Biotek Inc., Winooski, VT, USA).

### 2.3.4. Statistical Analyses

Statistical analyses were conducted using IBM SPSS Statistics for Windows (IBM Corp., 1968. Version 25.0, Armonk, NY, USA) and SAS (SAS Institute Inc., 1976. Base SAS® 9.4. SAS Institute Inc., Cary, NC, USA) software. Data was analyzed via one-way ANOVA to distinguish differences between treatments at each sampling time. Repeated measures (two-way ANOVA) were run to determine significant interactions between individual parameters and sampling time for appropriate groupings. Duncan's post hoc test was used to distinguish differences and considered significant at $p < 0.05$.

## 3. Results and Discussion

### 3.1. Background Soil Characteristics, Climatic Conditions, and Cover Crop Contributions

Soil at the experimental site was a sandy loam (73% sand, 17% clay, and 10% slit) with an average pH of 7.6. The experiment was conducted at a certified-organic farm where the average SOM content was ~17%, fairly high compared to the mineral soils (average < 2%; [24]) in the same region. Additionally, at the start of the experiment, inorganic N levels were also higher than expected for the area at 34.24 g kg$^{-1}$. These values are unusual for soils within the Redland area and can be explained by land use practices by the farm manager. Prior to carambola being planted, mature sapodilla trees were growing in this area. These trees were treated with the same 5–3–2 poultry manure (N–P–K) as used in our experiment at a rate of ~224 N kg ha$^{-1}$ per year, nearly double the amount added to juvenile carambola for our study. Mature trees also contribute leaf litter, which can add supplemental OM and N to soil as it decomposes [47]. As a result, these cultural practices had a great impact on soil health parameters and as such were reflected through enhanced OM and nutrient content within the soil at the start of the experiment and throughout season 1.

The average air and soil temperatures were 26.4 °C and 28.8 °C, respectively, for summer and 21.4 °C and 23.7 °C, respectively, for winter months (Figure 2). Precipitation trends were highest from July to August (18 to 23.7 cm average) in both years (Figure 2), as expected for South Florida, given climatic trends that result in wet summers and dry winters [37]. As such, a reduction in precipitation can be observed beginning in the fall months (September to November) and continuing throughout the year until summer. The first-year cover crop growing season (May 2018 to August 2018) received 64% higher rainfall than the second season (May 2019 to August 2019). Relative humidity (%) remained consistent (range 81 to 86%; average ~83% per month) throughout the experiment except for March 2019 to May 2019, when increased temperature and low rainfall resulted in lower humidity (average 78%).

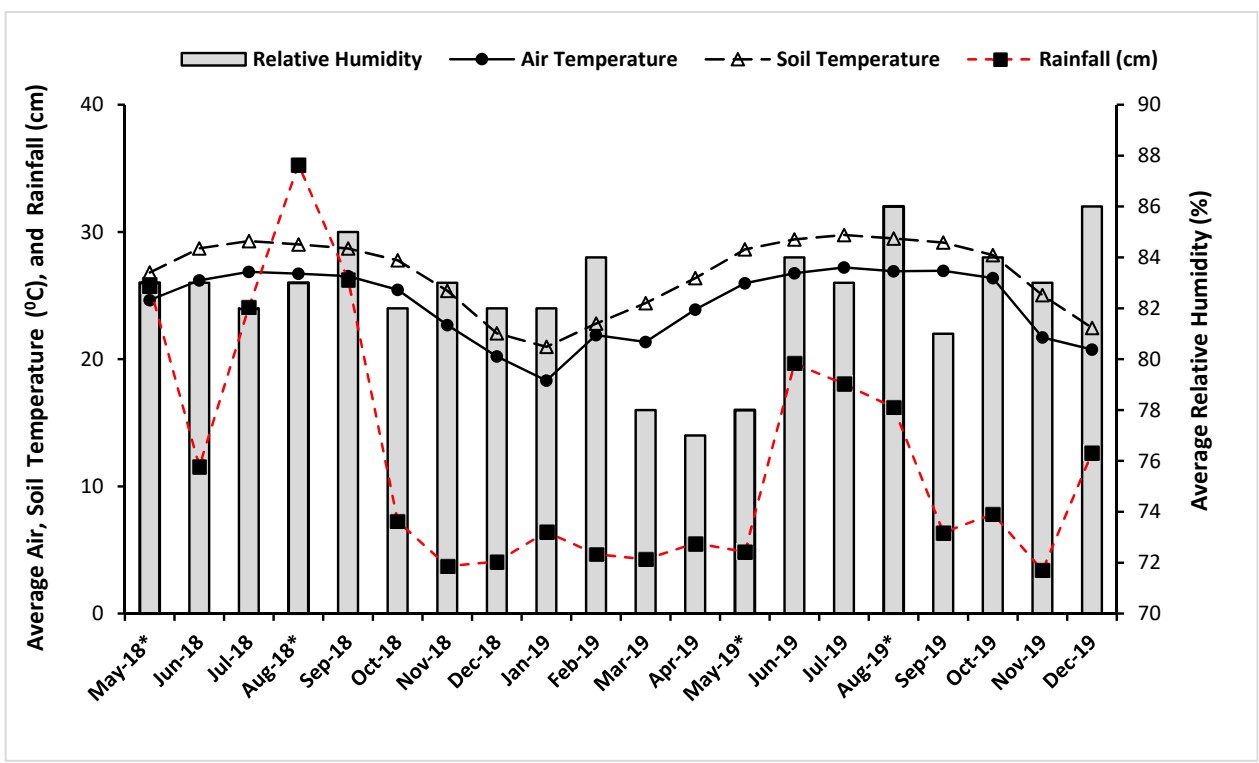

**Figure 2.** Climatic conditions of the sampling sites over the 1.5-year trial period. This graph represents average relative humidity (%), air temperature (°C), soil temperature (°C), and rainfall (cm). Months with (*) indicate cover crop planting in May and termination in August of each season.

Our results indicate that SH (with or without manure) produced significantly higher ($p < 0.05$) dry shoot biomass (range 10.7 to 5.4 Mg ha$^{-1}$) than VB (with or without manure; range 5.0 to 1.9 Mg ha$^{-1}$) for both growing seasons (Table 1). Consequently, SH biomass produced 33% more C than VB treatments throughout the experiment [40]. As a non-wood fiber crop, the SH stem can become strong and woody in its later growth stages, providing higher C additions over time [48]. As such, on average throughout our experiment, SH leaf material had a C:N ratio of 11.05, while stem material had a C:N of 32.44, which explains the higher carbon inputs to the system. Additional information on carbon inputs and accumulation related to cover crop biomass from this study can be found in our previously published work [40].

**Table 1.** Above-ground cover crop biomass, total N cover crop residue added to soil throughout season 1 and season 2, and total weed biomass. Three cover crop treatments (F = fallow, SH = sunn hemp, and VB = velvet bean) and three manure treatments (FM = fallow + poultry manure, SHM = sunn hemp + poultry manure, and VBM = velvet bean + poultry manure) were analyzed for these parameters. Values within a column followed by different letters denote statistical difference at $p < 0.05$ within the same season.

| | Season 1 | | | Season 2 | | |
|---|---|---|---|---|---|---|
| Treatment | Biomass (Mg ha$^{-1}$) | N (kg ha$^{-1}$) | Weed Biomass (kg ha$^{-1}$) | Biomass (Mg ha$^{-1}$) | N (kg ha$^{-1}$) | Weed Biomass (kg ha$^{-1}$) |
| F | - | - | 9475 a | - | - | 6496 a |
| FM | - | - | 7802 a | - | - | 6531 a |
| SH | 8.8 a | 177 ab | 1778 b | 9.4 a | 213 a | 1930 b |
| SHM | 10.7a | 238 a | 2322 b | 5.4 b | 135 b | 3302 b |
| VB | 4.5 b | 88 c | 3928 b | 2.7 c | 84 bc | 2742 b |
| VBM | 5.0 b | 136 bc | 2384 b | 1.9 c | 65 c | 3156 b |

When comparing total N contributed by cover crop dry matter, the SH treatments contributed up to ~92% (238 kg ha$^{-1}$) more biomass N than VB (88 kg ha$^{-1}$) treatments in season 1 ($p < 0.05$). Sunn hemp treatments produced up to 106% more biomass N than VB treatments ($p < 0.05$) in season 2. The difference in N accumulation by cover crops was a direct result of biomass production. There was a large variation in biomass production between SH and VB, likely attributable to their growth habits. Once SH is established, it grows and develops vertically, which is conducive to producing high biomass in confined spaces. Velvet bean produces large quantities of biomass through its vining growth habit and spreading surface roots [49]. As our experimental growing area around the carambola trees was limited to 8.8 m$^2$, VB produced less biomass than it would have if grown in a typical vegetable field setting. When comparing biomass production in a tomato production setting, Wang et al. (2009) [38] found that SH and VB produced similar quantities of biomass for both growing seasons in a 2-year study within the same region. However, when comparing SH and VB in a potted study within the RAA, Wang et al. (2006) [50] found that VB produced less biomass and TC when compared to SH for two consecutive growing seasons, findings that align with our study.

In addition to cover crop biomass rates, Table 1 reports weed biomass in the form of shoot dry matter production. The results reveal that weed biomass was up to 81% higher in the fallow plots where cover crops were not planted (F and FM). Weed biomass quantities reflect the ability of SH and VB to suppress weed growth and proliferation, a common environmental benefit of CC usage [51]. In general, CCs and weeds accumulated greater biomass in season 1 when compared to season 2, as reflected by shoot dry matter quantity (Table 1). Increased biomass in season 1 may have been the result of higher rainfall amounts during the seeding and germination period (May 2018) when compared to season 2 (April 2019) and throughout growing season 1 overall (Figure 2). Additionally, legumes tend to be more successful in non-amended soil. This phenomenon occurs because their nitrogen-

fixing capacity is enhanced, correlating directly with plant productivity and, in turn, dry biomass production [52]. This is a possible explanation for lack of CC biomass production in season 2, particularly for SHM and VBM which were treated with PM throughout the previous season.

### 3.2. Effect of Cover Crops on Soil Chemical Properties

Precise CC management has been shown to improve soil quality through increasing soil organic matter (SOM), providing nutrients to cash crops, and enhancing overall physical, chemical, and biological properties without the addition of synthetic inputs [53]. In our experiment, when considering soil pH within treatments, FM plots had significantly lower pH than all other treatments, remaining slightly alkaline throughout the experiment (7.75–7.83, $p < 0.05$, Table 2). This is likely the result of PM being added directly to the soil surface without the hindrance of CC residue. Conversely, the F plots had the highest pH consistently throughout the experiment (8.03–8.08, $p < 0.05$, Table 2), which can be attributed to the lack of OM residue added to the soil. All CC treatments remained similar to one another within the moderately alkaline range for both seasons, fluctuating from 7.85–8.03 ($p > 0.05$). Organic matter addition from the CCs may have played a role in acidifying soil in the short-term but was not reflected by data grouped by season. Calcareous soils generally have a high pH-buffering capacity; however, long-term fertilization can reduce the calcium carbonate ($CaCO_3$) content of naturally calcareous soils, ultimately lowering the buffering capacity [54]. While there were not obvious changes in soil pH throughout the study period, it is possible that over time, the loss of $CaCO_3$ and the increase of SOM could result in the acidification of soil with cover crop and manure additions.

**Table 2.** Displays average soil pH, organic matter percentage (SOM%), total carbon (TC), total nitrogen (TN), and C:N ratios throughout the experiment (n = 40 for each treatment). Three cover crop treatments (F = fallow, SH = sunn hemp, and VB = velvet bean) and three manure treatments (FM= fallow + poultry manure, SHM = sunn hemp + poultry manure, and VBM = velvet bean + poultry manure) were analyzed for these parameters. Values within a column followed by different letters denote statistical difference at $p < 0.05$.

| | pH | | SOM% | | TC (g kg$^{-1}$) | | TN (g kg$^{-1}$) | | C:N (mol:mol) | |
|---|---|---|---|---|---|---|---|---|---|---|
| Treatment | Season 1 | Season 2 | Season 1 | Season 2 | Season 1 | Season 2 | Season 1 | Season 2 | Season 1 | Season 2 |
| F | 8.03 a | 8.08 a | 15.00 bc | 13.99 b | 158.63 c | 159.81 c | 7.08 b | 7.10 bc | 26.55 a | 26.88 ab |
| FM | 7.75 b | 7.83 b | 17.86 a | 16.45 a | 183.63 a | 181.37 a | 9.17 a | 8.31 a | 23.38 a | 24.93 b |
| SH | 7.98 ab | 8.03 ab | 17.32 ab | 15.64 a | 175.32 ab | 169.67 abc | 8.95 a | 7.86 ab | 24.27 a | 25.18 ab |
| SHM | 7.85 ab | 7.99 ab | 17.45 a | 15.48 ab | 188.05 a | 170.79 abc | 9.25 a | 7.41 abc | 24.55 a | 27.37 ab |
| VB | 7.95 ab | 7.93 ab | 16.31 abc | 14.14 b | 178.11 ab | 173.89 ab | 7.70 ab | 6.55 c | 26.52 a | 28.10 a |
| VBM | 7.87 ab | 7.87 ab | 14.70 c | 15.48 ab | 166.42 bc | 168.94 bc | 7.89 ab | 6.98 bc | 25.38 a | 24.91 b |

Soil treated with FM and SHM accumulated the highest SOM% (17.45–17.86%, $p < 0.05$, Table 2), while the F and VBM (14.70–15%) treatments exhibited the lowest SOM% within season 1. Total carbon results also reflected this trend for season 1 which aligns with high input of aboveground shoot biomass and PM additions to the soil for the SHM treatment and direct addition of PM to the soil surface as applied in the FM treatment (Tables 1 and 2). The significant contribution of OM to the soil by SH treatments suggests that SH was effective for rapid addition of SOM during season 1, when CC biomass production was highest. Relatively lower SOM% and TC is expected for the F treatment with no manure or CC addition, and that treatment yielded consistently less C throughout the experiment when compared to other treatments. The comparably lower SOM% and TC within the VBM treatment throughout the experiment is likely attributed to a lesser contribution of cover crop biomass (as compared to SH treatments) along with the possibility of high decomposition/volatilization rates resulting from no-till management [55].

Soil organic matter results varied between treatments during season 2, as FM (16.45%) and SH (15.64%) had the highest SOM%, with that of SHM being marginally lower (15.48%,

$p < 0.05$, Table 2). This was also reflected in high soil TC levels for FM when compared to CC treatments. As previously mentioned, it is possible that because FM plots received manure treatments without the hindrance of CC mulch, change in SOM was more obvious in the short-term. Additionally, plots without CCs planted had a significantly higher degree of weed establishment throughout the experiment (Table 1). Mowing of weeds around plots occurred approximately every three months, making it possible for decaying weeds to contribute to soil building and nutrition for both fallow treatments. Fallow land within the F and FM treatments was advantageous for weed growth throughout the study, which may have contributed carbon to the soil through above- and belowground activities. High SOM values in SH plots are based on large contributions of CC biomass during season 2 growth (9.4 Mg ha$^{-1}$, $p < 0.05$). Overall, in regard to CC treatments, SH- and VB-treated soil were similar in TC to each other, yet lower than the FM treatment for season 2, which may have been a result of reduced CC biomass production from season to season.

Soil TN results from season 1 indicated that soil treated with F (7.08 g kg$^{-1}$) had the lowest TN. The FM, SH, and SHM treatments had the highest TN ($p < 0.05$, 9.17, 8.95, 9.25 g kg$^{-1}$, respectively), which corresponds with N inputs from biomass (Table 1). Soil TN was highest within soils treated with FM (8.31 g kg$^{-1}$) and lowest in those treated with VB (6.55 g kg$^{-1}$), with all other treatments being similar during season 2. Total N results for VB are likely a reflection of less successful biomass contribution by VB treatments during the second growing season (Table 1). These soil TN findings are consistent with other studies that have shown VB treatments add lower quantities of TN to soil than do SH treatments within the RAA subtropical climate [50,56]. Generally, although not statistically significant, in most cases ($p > 0.05$), a trend can be seen throughout both seasons in which treatments that received PM reflected higher TN.

Throughout season 1, there was no significant difference between treatments for soil C:N ratios ($p > 0.05$). Season 2 showed significantly lower C:N ratios for soils treated with FM (24.93) and VBM (24.91), indicating an environment more conducive to OM breakdown and nutrient cycling by microorganisms when compared to other treatments [57] (Eiland et al., 2001). The FM treatment consistently had the highest contributions of TC and TN to the soil throughout both growing seasons, which coincides with these results.

### 3.3. Soil Inorganic Nitrogen

Soil nutrient cycling, specifically nitrogen mineralization, is critical to crop health, as plant metabolism and vital processes are fueled by uptake of N [58]. When utilizing legume CCs for the purpose of green manure, residue added to the soil after termination adds N as a food source for soil organisms and subsequent crops [59].

When comparing soil inorganic N in the form of ammonium ($NH_4^+$) + nitrate ($NO_3^-$) + nitrite ($NO_2$) throughout the experiment, significant variability was observed at numerous sampling points and between treatments (Table A1, Figure 3). Soil inorganic N was highest, regardless of treatment, at the October 2018 (38.26 g kg$^{-1}$) sampling time and lowest in September 2019 (4.03 g kg$^{-1}$, $p < 0.05$). These results show a trend in which, two months after cover crop termination, inorganic N increases; this can also be observed in season 2, where N increases from September 2019 to October 2019 (Table A1, $p < 0.05$). During both growing seasons, air and soil temperature remained elevated from August–September during termination and throughout the first month of organic matter decomposition. These climatic conditions are characteristic of South Florida, as significant rainfall and high temperatures are conducive to organic matter decomposition and N mineralization [60]. Furthermore, the few months after CC termination are crucial for decomposition and N mineralization, which is apparent, as CC treatments showed significantly higher N at these time points.

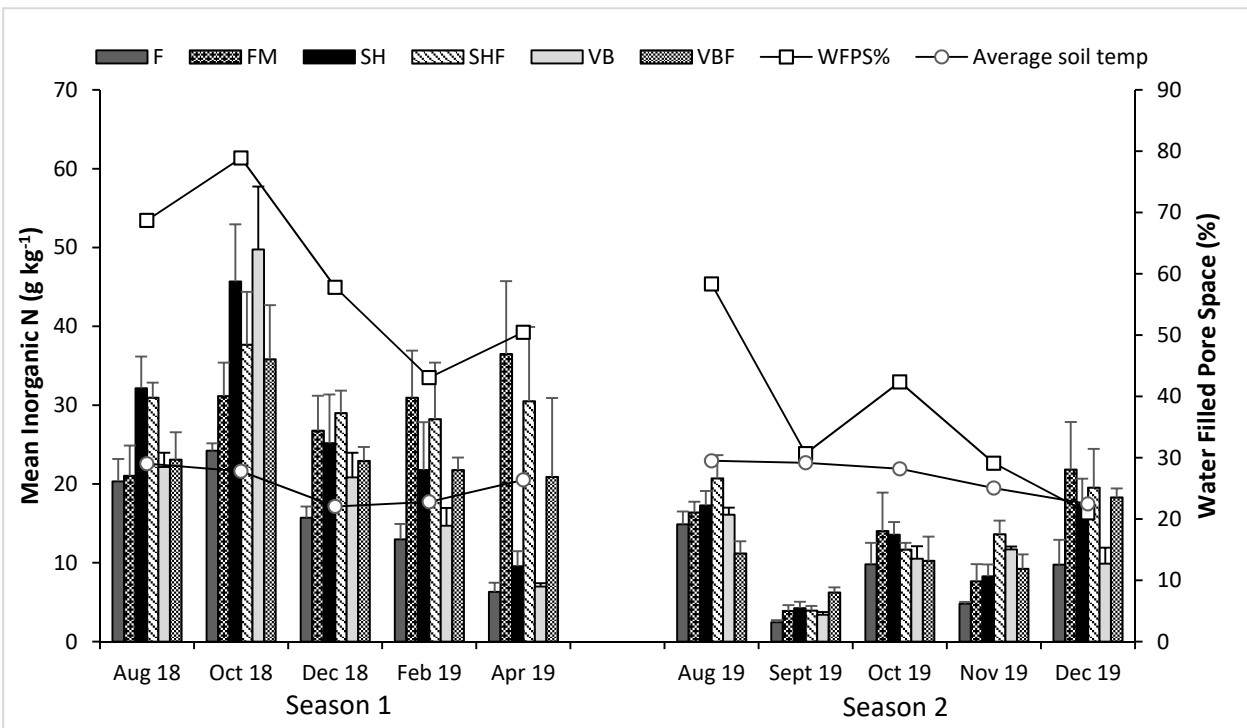

**Figure 3.** Displays average soil inorganic N concentrations by season (n = 4 for each treatment at each time point) with ambient soil temperature and WFPS. Three cover crop treatments (F = fallow, SH = sunn hemp, and VB = velvet bean) and three manure treatments (FM= fallow + poultry manure, SHM = sunn hemp + poultry manure, and VBM = velvet bean + poultry manure) were analyzed for these parameters. Error bars denote standard error.

At the September 2019 sampling period (1 month after the second CC termination), the average inorganic N was low, corresponding with lower water-filled pore space (WFPS)% as compared to other sampling periods (Figure 3). Water-filled pore space played a significant role in the N mineralization process throughout the experiment. In general, it was observed that as WFPS% increases, so does available N content and vice versa (Figure 3). Soils under no-till management are typically less aerobic and can have higher WFPS than those that undergo traditional tillage [61]. Tillage practices have a large influence on $N_2O$ emissions, which are generally higher in soil under no-till management, as anaerobic conditions are more common for such soil [62]. A water-holding capacity around 60% is the threshold for maximum aerobic activity ideal for ammonification and nitrification [63]. Plant-available N can be compared with WFPS, as previous studies have shown a link between WFPS, soil moisture, and $N_2O$ emission [64–66]. Results indicated that inorganic N content was elevated even when WFPS surpassed the 60% threshold at the October 2018 sampling time, although generally, high inorganic N was observed with WFPS at ~60%, ideal conditions for this parameter. This finding suggests that anerobic bacteria may have played a role in nutrient cycling throughout the study when WFPS% was high.

There are two groups of organisms responsible for N transformations in soils: ammonia-oxidizing bacteria (AOB) and ammonia-oxidizing archaea (AOA) [67,68]. The population size and response of AOA and AOB are highly related to soil type and management strategies. Traditionally, it has been found that AOB are more likely to contribute N additions in agricultural soils, as their populations are generally more elevated when N supply is higher, enhancing nitrification potential, while AOA are more commonly dominant in soils from more natural or diverse ecosystems [69]. Shen et al. (2008) [70] compared the abundance of AOB and AOA communities in an alkaline sandy loam (similar to the tested soil type) with various fertilizer treatments. They found significantly higher communities of AOB in soils treated with traditional N fertilizer when compared to organic manure treatments,

which is possibly explained by competition with heterotrophic bacteria, commonly present in soils amended with carbon (green or organic manures) [71]. Although not measured in this study, it is possible that an increased presence of AOA contributed significantly to N cycling in the present experiment, as these organisms are highly adaptable to extreme environmental conditions like low oxygen levels and are more common in diversified soils [72]. Because additions of OM to soil are favorable for microbial diversity, it is likely that the combination of CC inputs, paired with occasional anaerobic soil conditions, created a diversified microbial environment that facilitated N fixation and mobilization.

While WFPS may have played a role in N mineralization, it is probable that BNF had a significant impact on soil inorganic N content at the time of CC termination. At the August 2018 termination time, all soils treated with CCs had significantly higher inorganic N ($p < 0.05$) than the F soils (Table A1). Specifically, both SH and SHM exhibited the highest levels (32.15 and 30.94 g kg$^{-1}$, respectively). This sampling time is specifically interesting because this was before any PM fertilizer was applied, indicating successful N fixation due to the legume treatments. Legume symbiosis with rhizobium bacteria works to reduce $N_2$ to $NH_{4+}$ and $NO_{3-}$ in ideal climatic soil conditions [73]. Biological nitrogen fixation is dependent on many factors and can vary by species and effectiveness of rhizobium type/inoculation success [74]. Nezomba et al. (2008) [75] found that Crotalaria spp. had high potential to fix N in sandy soil. Specifically, Crotalaria juncea (SH) was estimated to have a 90% N fixation rate, resulting in 58 kg ha$^{-1}$ N provided to soil. Within our experiment, both SH and VB seeds were inoculated with the recommended cowpea-type rhizobium before planting to encourage nodulation. Genus *Crotalaria* (SH) has been shown to create symbiotic relationships with many strains of rhizobium bacteria, resulting in high potential for N fixation and biomass accumulation [76]. Conversely, much less is known about the genus *Mucuna* (VB) and its rhizobial-host-plant interactions. Cowpea-type rhizobium is compatible with genus *Mucuna*; however, successful nodulation has not been shown with a wide variety of rhizobium species [76]. These factors may have had an impact on the differences in N content in soil between the SH and VB treatments, specifically during and directly after the CC growing seasons (August 2018 and August 2019) in which SH-treated soil showed more success in providing available N.

In addition to termination, there were multiple sampling times in which significant differences in soil inorganic N were observed between treatments. At the October 2018 sampling time, two months after CC termination, VB-treated soil had higher inorganic N (49.74 g kg$^{-1}$, $p < 0.05$) than the F treatment, with all other treatments being similar to one another. The VB treatment also received higher biomass contribution in season 1 than VBM (~9% higher), which may have contributed to significantly higher levels of inorganic N. It is apparent that at this sampling time (two months after termination and first PM application), N from CC residue and PM application was in various stages of decomposition, and N was being utilized by the carambola tree, as it was readily available. Generally, we saw that VB/VBM residue had a significantly lower C:N ratio than SH [40], which enhanced N cycling from VBM treatments soon after CC incorporation. This pattern can be seen again at the October 2019 sampling time, where the VBM-treated soils had significantly higher levels of inorganic N than all other treatments (6.23, $p < 0.05$), which was a reflection of PM addition and the low C:N ratio of the residue contributing to higher levels of N mineralization.

At the December 2018 sampling time, soils treated with SHM (29.00 g kg$^{-1}$, $p < 0.05$) were higher in inorganic N than the F-treated soils (15.71 g kg$^{-1}$), with all other treatments being statistically similar to one another. This result also corresponds to season 2 results at the November 2019 sampling time in which SHM (13.63, $p < 0.05$) soils showed significantly higher N than all other treatments (Table A1). These results indicate that, 3 to 4 months after cover crop termination, plots that received SH and PM combined had the greatest capacity for N mineralization. This corresponds with the findings of Rao and Li (2003) [60], who observed that SH had the highest level of cumulative N mineralization

in calcareous Krome soil at 16 to 20 weeks after CC residue was added at ambient South Florida temperature conditions.

At six months (February 2019) and eight months (April 2019) after termination, the impact of CC residue on soil inorganic N begins to taper off, and added PM becomes the main source of plant-available N. At six months after termination (February 2019), the FM-treated soils (30.94 g kg$^{-1}$, $p < 0.05$) had the highest levels of inorganic N compared to F-treated soil (12.96 g kg$^{-1}$), with all other treatments being similar to one another (28.23–14.71 g kg$^{-1}$). However, at the April 2018 sampling time, eight months after termination, a distinct difference can be seen between treatments that received PM and those that did not. Figure 3 shows this distinct phenomenon, in which FM, SHM, and VBM had higher levels of inorganic N (36.49, 30.50, and 20.90 g kg$^{-1}$, respectively, $p < 0.05$) when compared to treatments that did not receive any PM additions. These results suggest that at six to eight months after termination, CC residue ceases to contribute to N availability in the soils at our research site.

### 3.4. Soil Enzyme Activity

We chose to study two enzymes, β-1-4-glucosidase, which is associated with C cycling, and β-N-acetylglucosaminidase, which is associated with N cycling [77]. Soil β-1-4-glucosidase rates differed over time and between treatments (Figure 4). Throughout the sampling times, with all treatments considered, the August 2019 time (second CC termination) had the highest rates of β-1-4-glucosidase. There was no significant difference between treatments at individual sampling times except for the August 2018 (the end of cover crop growing season 1) and August 2019 (the end of cover crop growing season 2). At both August times, the SHM treatment had the highest β-1-4-glucosidase level (0.0386 and 0.1393 μmol m 2 s, respectively), with the levels of all other treatments significantly lower ($p < 0.05$), which may be indicative of sunn hemp excreting carbon into the rhizosphere, enhancing overall microbial diversity. Overall, soils treated with SHM reflected the highest numeric β-1-4-glucosidase, although no significant difference was discernable ($p > 0.05$, Figure 4).

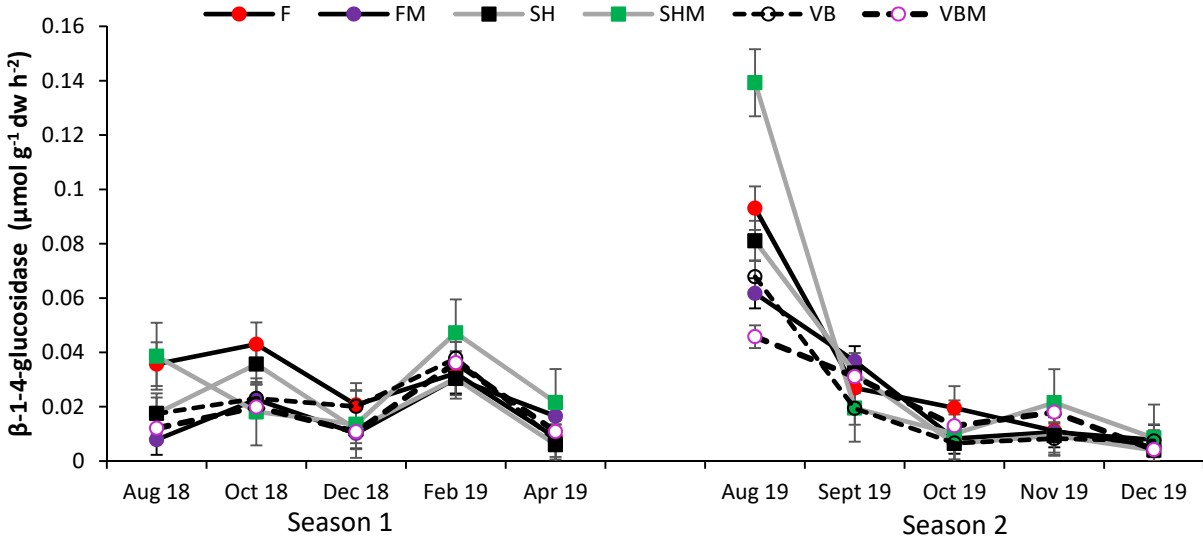

**Figure 4.** Displays average soil β-1-4-glucosidase rates (n = 4) over a two-season period after cover crop growth and termination. The lines at each sampling time represent three cover crop treatments (F = fallow, SH = sunn hemp, and VB = velvet bean) and three manure treatments (FM= fallow + poultry manure, SHM = sunn hemp + poultry manure, and VBM = velvet bean + poultry manure). Error bars denote standard error.

The enzyme β-N-acetylglucosaminidase reflects N cycling by microbial biomass and overall breakdown of OM [78]. Soil β-N-acetylglucosaminidase results indicated change

over time, regardless of treatment. Like β-1-4-glucosidase, the highest occurrences were in August 2019 (0.0094 μmol m 2 s), at the second CC termination (Figure 5). There was no significant difference in β-N-acetylglucosaminidase between treatments at any of the individual sampling times except for August 2019. At the August 2019 time, soils treated with SHM (0.0169 μmol m 2 s) showed significantly higher β-N-acetylglucosaminidase activity than the rest of the treatments (Figure 5).

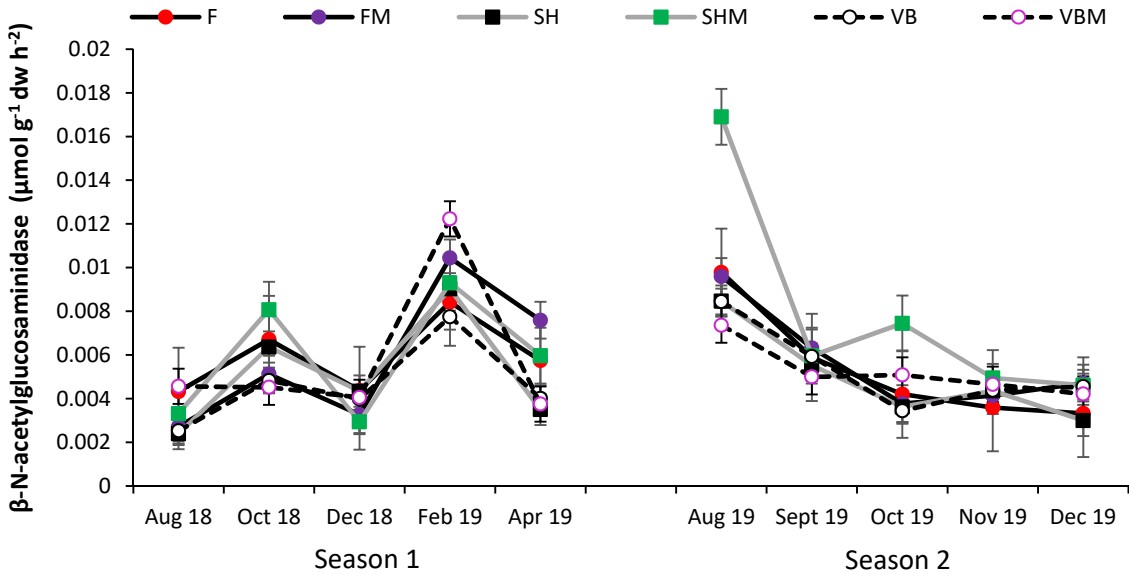

**Figure 5.** Displays average β-N-acetylglucosaminidase rates (*n* = 4) in soil over a two-season period after cover crop growth and termination. The lines at each sampling time represent three cover crop treatments (F = fallow, SH = sunn hemp, and VB = velvet bean) and three manure treatments (FM= fallow + poultry manure, SHM = sunn hemp + poultry manure, and VBM = velvet bean + poultry manure). Error bars denote standard error.

These enzyme activity results can be reflective of two processes. After the 90 days of the second growing season, it appears that the SHM treatment was effective in providing belowground stimulation to soil, and as such, the breakdown of glucose and transformation of N occurred, as shown by increased enzyme activity in this sampling period (August 2019). These results coincide with higher concentrations of inorganic N in the corresponding sampling period in which SHM had the highest levels (Figure 3, $p < 0.05$). A study conducted by Maltais-Landry (2014) [79] concluded that legume CCs had high β−glucosidase activity in the rhizosphere by the end of their growing season when compared to non-legumes, which was especially apparent when legume cover crops were combined with composted poultry manure. Our study shows that this is true in a no-till field setting for SH treated with PM (SHM). Shoot and root contributions supplied by cover crops are specifically important in no-till systems, as root exudates added during the growing season and organic material added after termination stimulate soil microbial communities [80].

## 4. Conclusions

Cover crops have rarely been explored as a soil management method for tropical fruits. This study shows that tropical leguminous cover crops have potential as beneficial soil amendments to add OM and nutrients and promote nutrient cycling by stimulating microbial activity. This experiment was conducted in an organic production farm where cover crops were intercropped with young carambola trees. Carambola trees require ~90–270 N kg ha$^{-1}$ per year [41]. With the seeding rate utilized in this experiment, SH and VB treatments potentially provided sufficient amounts of total dry matter N to supply carambola trees with the N that they require. The results suggest that SH treatments

produced the greatest amount of dry biomass material, ranging from 48 to 71% higher than VB treatments over two growing seasons. Consequently, SH also accumulated significantly higher amounts of TC and TN within its dry biomass that was added to the soil. Sunn hemp, SHM, and FM treatments showed the greatest accumulation of soil organic matter, TC, and TN. Soil inorganic N fluctuated throughout the experiment. Our results indicate that, generally, VB-treated soils had their highest available N around 2 months post termination, while SH-treated soils exhibited significantly higher N values at CC termination time. Sunn hemp + PM treatments had highest soil N availability around 4 months after CC termination. Soil enzyme activity results indicate that at CC termination, SHM exhibited the highest β-1-4-glucosidase and β-N-acetylglucosaminidase levels among all treatments.

With all results considered, SH and VB both have the potential to act as soil enhancers for fruit production in tropical and subtropical settings. Applying the findings to tropical fruit production, these cover crops can provide chemical and biological benefits to enhance soil for the successful growth of tropical fruit trees. These cover crops can be utilized in combination with PM or other organic fertilizers for ideal crop development and soil improvement. With the growing issues of soil erosion and organic matter depletion, it is imperative that farmers consider these matters and incorporate management strategies that ensure the long-term sustainable productivity of their land.

**Author Contributions:** Conceptualization: A.F., Y.L. and K.J.; methodology: A.F., Y.L. and K.J.; formal analysis: A.F.; investigation and data curation: A.F. and S.D.; resources: K.J.; writing—original draft preparation: A.F.; visualization: A.F. and S.D.; writing—review and editing: A.F., S.D., Y.L. and K.J.; funding acquisition: K.J. All authors have read and agreed to the published version of the manuscript.

**Funding:** This research was funded through FIU's Agroecology Program via the U.S. Department of Agriculture–National Institute of Food and Agriculture, National Needs Fellowship (2015-38420-23702), FIU Tropics and the Susan S. Levine Trust, and the FIU Graduate School Doctoral Evidence Acquisition and Dissertation Year Fellowships.

**Institutional Review Board Statement:** Not applicable.

**Informed Consent Statement:** Not applicable.

**Data Availability Statement:** Not applicable.

**Acknowledgments:** We would like to acknowledge Marc Ellenby and family for supplying the field site for this experiment.

**Conflicts of Interest:** The authors declare no conflict of interest.

## Appendix A

**Table A1.** Displays average soil inorganic N concentrations at each sampling time (*n* = 4 for each treatment at each time point). Three cover crop treatments (F = fallow, SH = sunn hemp, and VB = velvet bean) and three manure treatments (FM= fallow + poultry manure, SHM = sunn hemp + poultry manure, VBM = velvet bean + poultry manure) were analyzed for these parameters. Different uppercase letters denote statistical difference between sampling times, different lowercase letters denote statistical differences between treatments at each individual sampling time ($p < 0.05$).

| | Soil Inorganic N (g kg$^{-1}$) | | | | | | |
|---|---|---|---|---|---|---|---|
| Season 1 | Overall | F | FM | SH | SHM | VB | VBM |
| August 2018 | 25.48 B | 20.32 c | 21.04 bc | 32.15 a | 30.94 ab | 22.16 bc | 23.08 abc |
| October 2018 | 38.26 A | 24.22 b | 31.15 ab | 45.70 ab | 37.62 ab | 49.74 a | 35.82 ab |
| December 2018 | 23.42 B | 15.71 b | 26.76 ab | 25.18 ab | 29.00 a | 20.83 ab | 22.93 ab |
| February 2019 | 20.90 B C | 12.96 b | 30.94 a | 21.76 ab | 28.23 ab | 14.71 ab | 21.77 ab |
| April 2019 | 17.58 C | 6.32 b | 36.49 a | 9.55 b | 30.50 a | 6.97 b | 20.90 ab |

**Table A1.** *Cont.*

| | | | | | | | |
|---|---|---|---|---|---|---|---|
| | Soil Inorganic N (g kg$^{-1}$) | | | | | | |
| **Season 2** | | | | | | | |
| August 2019 | 16.30 CD | 14.89 ab | 16.39 ab | 17.28 ab | 20.70 a | 16.10 ab | 11.19 b |
| September 2019 | 4.03 F | 2.45 b | 3.90 b | 4.23 b | 3.94 b | 3.41 b | 6.23 a |
| October 2019 | 11.54 DE | 9.81 a | 14.02 a | 13.56 a | 11.67 a | 10.51 a | 10.24 a |
| November 2019 | 9.25 EF | 4.82 c | 7.65 bc | 8.27 bc | 13.63 a | 11.69 ab | 9.24 abc |
| December 2019 | 16.36 CD | 9.76 a | 21.83 a | 17.43 a | 19.52 a | 9.87 a | 18.30 a |

**Table A2.** Displays bivariate correlations for recorded parameters throughout the study.

| | SOM [a] | TC [b] | TN [c] | C:N | Inorganic N [d] | MUFN [e] | MUFC [f] | Moisture [g] |
|---|---|---|---|---|---|---|---|---|
| **pH** | −0.326 ** [i] | −0.165 * | −0.315 ** | 0.299 ** | −0.511 ** | −0.273 ** | −0.171 ** | −0.240 ** |
| **SOM** | | 0.690 ** | 0.764 ** | −0.541 ** | 0.506 ** | 0.081 | 0.130* | 0.294 ** |
| **TC** | | | 0.740 ** | −0.353 ** | 0.404 ** | −0.034 | 0.060 | 0.234 ** |
| **TN** | | | | −0.783 ** | 0.532 ** | 0.055 | 0.124 | 0.410 ** |
| **C:N** | | | | | −0.346 ** | −0.101 | −0.141 | −0.268 ** |
| **Inorganic N** | | | | | | 0.243 ** | 0.212 ** | 0.613 ** |
| **MUFN** | | | | | | | 0.430 ** | 0.313 ** |
| **MUFC** | | | | | | | | 0.214 ** |

[a] Soil organic matter. [b] Soil total carbon. [c] Soil total nitrogen. [d] Soil inorganic nitrogen. [e] β-N-acetylglucosaminidase. [f] β-1-4-glucosidase. [g] Soil moisture. [i] Representing Pearson's correlation coefficient (r) significant at $p \leq 0.05$ (*) or $p \leq 0.01$ (**).

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
