# Peer review of "Influence of Leguminous Cover Crops on Soil Chemical and Biological Properties in a No-Till Tropical Fruit Orchard"

_land, doi:10.3390/land11060932_

Round 1
Reviewer 1 Report
Manuscript review:
Influence of Leguminous Cover Crops on Soil Chemical and Biological Properties in a No-till Tropical Fruit Production Setting
The aim this manuscript was to quantify the response of dynamic soil characteristics to cover crop incorporation in a young carambola grove by exploring responsive soil parameters.
The subject of the authors of the manuscript is very important due to the degradation of the soil environment in terms of organic matter. The authors undertook research on cover crops in terms of soil management. The paper is of interest to the readers of Land journal.
I am not a native English speaker, so I cannot undergo revision of the English style and grammar of the manuscript.
The article is well-written and easy to understand.
The manuscript does not conform to the editorial requirements:
"References should be numbered in order of appearance and indicated by a numeral or numerals in square brackets — e.g., [1] or [2,3], or [4–6]. References must be numbered in order of appearance in the text (including citations in tables and legends) and listed indi-vidually at the end of the manuscript. " Should be improved!
In Introduction: Information on soil enzyme activity is lacking.Why did the authors undertake the study of soil enzymatic activity?
There is no research hypothesis
Material and method
Has enzyme determination been performed on fresh or dry soil? Please put information in the text.
Why did the authors choose to determine the activity of β-1-4-glucoside and β-N-acetylglucosaminidase?
Please characterize the principle of action of these two enzymes in relation to macroelements in soil (C and N)
Line 235, 239 - use superscript with units
Why is the study not presented the analysis of the correlation between the studied soil parameters? Please justify or complete the statistical analysis
Author Response
Please refer uploaded Reviewer 1 response.

Reviewer 2 Report
Review for the manuscript entitled “Influence of Leguminous Cover Crops on Soil Chemical and Biological Properties in a No-till Tropical Fruit Production Setting”.
In this study, the authors aimed to examine the response of dynamic soil characteristics to cover crop incorporation in a young carambola grove by exploring responsive soil parameters. According to their results, sunn hemp and velvet bean both have potential to act as soil enhancers for fruit production in tropical and subtropical settings. The authors have done a proper experimental design, adequate laboratory and statistical analysis. The manuscript is very well written, and results are persuasive. Therefore, I recommend this manuscript for publication.
Suggestion:
It is suggested to make the conclusion shorter with a focus on the key conclusions of the manuscript
Author Response
Please refer uploade reviewer 2 response file

Reviewer 3 Report
I recommend that you also mention the laboratory or laboratories where all the analyzes were performed
Author Response
Please refer uploaded reviewer 3 response.

Reviewer 4 Report
General comments:
This study investigated the impacts of leguminous cover crops (Sunn hemp and Mucuna pruriens) on soil chemical (total carbon and total N) and biological properties in a no-till tropical fruit (young carambola) production setting based on field experiment in two seasons. Its research content and conclusions are of great significance to the sustainable development of local agriculture, and also provide valuable management reference for related regions. In addition, the writing of the manuscript is standardized and easy to read, and it is recommended to publish it after minor revisions.
Special comments:
Line79, the second “in” should be removed.
Line170, The resolution of the figure is low, and (A) and (D) are missing.
Line256, Need to point out what the squares and circles represent.
Line229, “3. Results” should change into “3. Results and discussion”, and increase related discussions to make the research more meaningful.
The figures throughout the manuscript should remain consistent, either in color (Figure 4 and Figure 5) or in black and white (Figure 2 and Figure 3).
Author Response
Please refer uploaded reviewer 4 response
